# Cough in Protracted Bacterial Bronchitis and Bronchiectasis

**DOI:** 10.3390/jcm13113305

**Published:** 2024-06-04

**Authors:** Hinse Wiltingh, Julie Maree Marchant, Vikas Goyal

**Affiliations:** 1Department of Respiratory and Sleep Medicine, Queensland Children’s Hospital, Brisbane, QLD 4101, Australia; hinsewiltingh@hotmail.com (H.W.); jm.marchant@qut.edu.au (J.M.M.); 2Australian Centre for Health Services Innovation, Queensland University of Technology, Brisbane, QLD 4000, Australia; 3Department of Pediatrics, Gold Coast Health, Gold Coast, QLD 4215, Australia

**Keywords:** cough, protracted bacterial bronchitis, bronchiectasis, children, review

## Abstract

Chronic cough in children is a common condition for which patients seek medical attention, and there are many etiologies. Of the various causes of chronic cough in children, protracted bacterial bronchitis (PBB) is one of the commonest causes, and bronchiectasis is one of the most serious. Together, they lie on different ends of the spectrum of chronic wet cough in children. Cough is often the only symptom present in children with PBB and bronchiectasis. This review highlights the role of cough as a marker for the presence of these conditions, as well as an outcome endpoint for treatment and research.

## 1. Introduction

Cough remains the most frequent reason for patients to seek medical attention in primary care [1]. This includes children with a chronic wet cough, i.e., a wet cough that persists for more than 4 weeks [2]. Chronic cough can affect children’s quality of life (QoL) by interfering with their sleep, social development, and education [3]. Furthermore, it can lead to increased parental stress due to worry about the cause and frequent doctor visits [4]. Defining the etiology of a child’s cough and initiating appropriate treatment is therefore important in decreasing the burden of disease. Two conditions that commonly present with a chronic wet cough in children and should be considered in the diagnostic process, are protracted bacterial bronchitis (PBB) and bronchiectasis [5].

PBB is now defined as the presence of an isolated chronic wet cough and resolution with a 2-week course of appropriate antibiotics [6]. It has been proposed that PBB and bronchiectasis represent different ends of a spectrum of endobronchial suppuration [7] and PBB can be a precursor to bronchiectasis [8]. Bronchiectasis is defined as abnormal airway dilatation seen on a chest computed tomography (CT) scan with associated symptoms, including chronic productive cough and recurrent respiratory exacerbations [9]. It causes significant morbidity in children, especially its recurring respiratory exacerbations that are associated with reduced QoL [10]. This review will highlight the role of cough as a marker for the presence of these conditions, recent developments in understanding and managing cough in PBB and bronchiectasis, and the use of cough as an outcome endpoint for treatment and research. We searched the PubMed and Cochrane databases for cough in protracted bacterial bronchitis bronchiectasis. We used the terms “protracted bacterial bronchitis” and “cough” for search, along with “bronchiectasis” and “cough”. We then searched using keywords “protracted bacterial bronchitis”, “bronchiectasis”, and “cough duration” in the same databases. Our searches were restricted to reports in English. We only focused on the studies describing cough characteristics in PBB and bronchiectasis. The last searches were conducted on 25 April 2024.

## 2. Chronic Wet Cough Spectrum

Together with chronic suppurative lung disease (CSLD), PBB and bronchiectasis form the “chronic wet cough spectrum” in children (Figure 1) [7]. The three conditions share many similarities, including the presence of a chronic wet cough, airway neutrophilia, and pathogenic airway bacteria [11]. However, based on clinical presentation and chest high-resolution computed tomography (c-HRCT) results, distinguishing features can be recognized. Children with PBB present with isolated wet cough, without the presence of other signs or symptoms, whereas children with CSLD present with additional signs and/or symptoms in the absence of bronchial dilatation on a c-HRCT [7]. Children with bronchiectasis may also have additional signs and/or symptoms and their c-HRCT may show abnormal dilatation of the airways, characterized by increased broncho–arterial ratios (BAR). The BAR is used is defined as the ratio of the inner diameter of an airway to the outer diameter of adjacent artery that is within 5 mm in a nontangential plane. Pediatric guidelines recommend using a BAR of >0.8 to define bronchiectasis (instead of the adult cutoff of >1 or >1.5) [12]. As the disease entities are part of the same spectrum, a child’s diagnosis can progress from one condition to the next, dependent on appropriate management. If treated appropriately, wet cough in PBB will resolve after a protracted course of antibiotics. On the contrary, without adequate treatment, children with PBB can progress to develop CSLD or bronchiectasis [7].

Though the exact underlying pathophysiological mechanism is not completely understood, there is likely an interplay of genetics, host’s environment, micro-organisms, and immune response which results in the development of PBB, and if it progresses, to CSLD or bronchiectasis. Cole’s vicious cycle hypothesis explains that an inadequate response to an infection results in bronchial wall damage [13]. This in turn leads to decreased airway clearance, resulting in further airway colonization and thereby inflammation, creating a vicious cycle [13]. In recent years a new model, the vicious vortex, has been proposed, in which these different factors influence each other simultaneously rather than sequentially [14].

## 3. What Is Protracted Bacterial Bronchitis?

PBB was first proposed as separate diagnosis in 2006 [6] in children with chronic wet cough without the presence of other localizing features, and presence of airway neutrophilia and response to antibiotic treatment [6]. PBB is now clinically defined as an isolated chronic wet cough with response to ≥2 weeks of antibiotics. With resolution of the cough, where no signs, symptoms, or “red flags” suggestive of an alternative etiology, a bronchoscopy is no longer required to make the diagnosis [2]. PBB is considered recurrent when >3 episodes/year occur [15]. Children with PBB have preserved systemic adaptive immunity with normal serum immunoglobulin levels and normal antibody-mediated responses to both protein (tetanus) and conjugated protein–polysaccharide-based (*H. influenzae* type b) vaccines, and do not have any other signs of systemic inflammation [16].

PBB, as a common cause of chronic wet cough in children, is increasingly being recognised [17,18,19], and it is now integrated in the pediatric chronic cough guidelines internationally, including the American [20], European [16], British [21], and Australian cough guidelines [22]. PBB can be seen at any age in children but the incidence is higher in younger children (mean age 1.8–4.8 years) [23], and is more commonly seen in male children attending daycare [24].

Children with PBB have a high burden of illness and impaired QoL. having seen multiple doctors prior to presentation (80% >5 doctor visits for cough) and tried many medications, in particular asthma medications [4,18]. QoL scores in children with PBB are comparable to children with bronchiectasis and chronic airway disease such as asthma [4]. The impaired QoL scores return to normal with treatment and cough resolution [25].

Lower airway specimens of children with PBB have shown the presence of airway neutrophilia combined with presence of bacteria as *H. influenzae*, *S. pneumoniae*, and *M. catarrhalis* [19]. Chest X-rays, if performed, are often normal but can show peri-bronchial changes. *H. influenzae* in airway samples of children with PBB can be a significant predictor for the development of bronchiectasis [26].

## 4. What Is Bronchiectasis?

Bronchiectasis is a heterogenous condition characterized by dilation of the airways and chronic airway suppuration. An Australian study reported an annual incidence of 735:100.000 in indigenous children of Central Australia [27]. In high income countries, incidence are substantially lower, ranging from 0.2:100.000 in the UK [28] and up to 2.3:100.000 in Ireland [29].

Even though the exact pathophysiology of bronchiectasis remains unknown, post-infectious bronchiectasis is the most common reported etiology in children [29]. In a literature review of 989 children from 12 studies with bronchiectasis, 63% had an identified predisposing condition (including 19% with PBB or other infections, 16% with primary immunodeficiency, and 10% with aspiration) [30]. However, the prevalence and nature of underlying conditions varies depending on the population studied and the intensity of the evaluation, like primary ciliary dyskinesia, congenital malformations, and aspiration [30,31,32,33].

Bronchiectasis can lead to high morbidity and even mortality, depending on its severity. Exacerbations are frequent, leading to distress for children and parents [3,34]. Symptoms may vary between individuals and are not necessarily related to the disease severity seen on a c-HRCT. The main symptoms include a chronic wet cough, sputum production, and recurring lower airway infections. Additional signs and symptoms can be digital clubbing, chest infections, dyspnea, fatigue, wheezing, and in severe cases, even hemoptysis [35]. Since symptoms can be non-specific, bronchiectasis is often misdiagnosed as a different respiratory disease like asthma resulting in mismanagement.

If bronchiectasis is identified earlier and proper treatment is utilized, the prognosis might be good. Conversely, when diagnosis is delayed and appropriate treatment is lacking, bronchiectasis can cause severe morbidity.

## 5. Cough in PBB

Wet cough >4 weeks is the hallmark of PBB. PBB is one of the most common causes of chronic wet cough in young children (<5 years of age), accounting for approximately 40 percent of referrals to pediatric pulmonary specialist clinics [36]. A significant correlation between duration of wet cough and severity of the radiographic findings on c-HRCT scans and the degree of inflammation has been demonstrated [37]. In adult non-smokers newly diagnosed with bronchiectasis, the duration of productive cough had a significant negative correlation with airway obstruction measured on spirometry (r = −0.51% predicted FEV_1_ per year of chronic cough, 95% CI −0.69, −0.33; *p* < 0.001), demonstrating that the symptoms start in childhood [38]. Thus, it has been advocated that children with chronic wet cough should be evaluated carefully for an underlying etiology [25,37]. If managed with an appropriate antibiotic course of two weeks, wet cough due to PBB resolves [39]. However, some patients require a longer treatment course of up to 4 weeks duration [40]. When longer treatment is required, an underlying cause must be taken into consideration [15].

Generally, with appropriate treatment the prognosis of PBB is regarded as good [23]. A 5-year prospective cohort study investigated the long-term data of children with PBB [26]. It showed that the percentage of children experiencing recurrence decreased over the years with a more than 3-fold reduction in PBB exacerbations during follow-up [26]. This study also demonstrated an overlap with asthma-like conditions; clinician-diagnosed asthma at final follow-up was present in 27.1% of children with PBB.

### 5.1. Response to Antibiotics in PBB

The ERS taskforce has now defined clinical PBB as a chronic wet cough, that responds to an antibiotic course of two or more weeks [16]. Since PBB is primarily a clinical diagnosis, with a cough being the most important symptom, the primary outcome to evaluate during treatment is resolution of the cough.

The treatment of choice for PBB is amoxicillin-clavulanate because of its activity against β-lactamase and because it is effective against the most frequently encountered pathogens: *H. influenzae*, *S. pneumoniae*, and *M. catarrhalis* [15]. A study comparing the effectiveness of two weeks of amoxicillin-clavulanate to placebo, reported a 48% cough resolution in the antibiotic group, compared to a 16% resolution rate in the placebo group [41]. Another study showed that more than half of the patients with PBB were cough free after two or more courses of antibiotics [18]. A systemic review examining the effectiveness of antibiotics in resolving cough concluded that there is high quality evidence that 2-weeks of antibiotics is, in general, sufficient in treating PBB [42]. However, it recognizes that a small group of children may require a prolonged antibiotic course. A retrospective study of 144 children with chronic wet cough, demonstrated that those who did not respond to at least 4 weeks of oral antibiotics were at significantly greater risk of having bronchiectasis ([OR_adj_] 20.9, 95% CI 5.4–81.8) [43].

The American cough guidelines recommend two weeks of treatment, followed by an additional two weeks, if the cough has not resolved [44]. Simultaneously, the European Respiratory Society recommends 2–4 weeks of antibiotics [16]. On the contrary, the British Thoracic Society (BTS) states that treatment consists of 4–6 weeks of antibiotics [21]. Nonetheless, the most optimal treatment length is yet to be determined. A retrospective study further investigated the association between treatment duration and recurrent relapse [45]. It showed 6-week vs 2-week course of antibiotics to be associated with a decrease in relapse, OR 0.115 (CI 0.026–0.509, *p* = 0.019).

A multicentre RCT comparing two weeks of antibiotics to four weeks in children with PBB, found no significant difference in the primary outcome between the two groups, as both had similar rates of cough resolution [40]. However, the 4-week group did have a significantly longer time till a next episode of PBB: 150 days vs. 38 days.

### 5.2. Role of Cough as an Outcome Measure in PBB Interventions

Since assessing outcomes varies from research centres and internationally, a study developed a core outcome set (COS) for PBB, to standardize outcome assessment [46]. A total of twenty outcomes were identified, of which six were included in the final COS. Of these six outcomes, resolution of cough has been identified as the most important outcome, along with relapse of chronic cough as the second most important outcome. A paediatric QoL instrument for children with chronic cough has been developed and validated, which measures the impact of cough symptoms on children’s well-being [3]. The cough score in children [47] has been validated as a clinical and research tool and resolution of cough identified as a marker of improvement in cough [40,47]. The cough score is a verbal category descriptive score (VCD) ranging from zero (no cough) to five (cannot perform most usual activities due to severe coughing) [47]. A prospective cohort study investigated if patients’ characteristics, clinical characteristics, or additional investigation results had a correlation with cough resolution [48]. This study revealed that a significant but mild relationship exists between cough duration and cough resolution. Where a cough duration of one additional month requires 1.02 more days of treatment with antibiotics, compared to children who have had a cough for 1 month in total.

## 6. Cough in Bronchiectasis

The most common symptom in children with bronchiectasis is chronic wet cough [49]. Failure of chronic wet cough to respond to four weeks of antibiotics predicts presence of underlying bronchiectasis (adjusted odds ratio [OR_adj_] 20.9, 95% CI 5.4–81.8) [43]. Children often present with a history of wet cough starting early in life and/or persisting for months or years before being referred to a respiratory clinic [6]. Indeed, many adults newly diagnosed with bronchiectasis have had symptoms since childhood and these adults have worse lung function and more severe radiological features than those with shorter duration of chronic cough [50,51].

Chronic wet cough might be the only symptoms present in children with mild bronchiectasis, but history of exertional dyspnea, recurrent wheezing poorly responsive to asthma treatment or recurrent lower respiratory tract infections, hemoptysis, digital clubbing, and/or chest wall deformity also increase the likelihood of underlying bronchiectasis [27,52,53,54,55,56,57]. Some children might have crackles on auscultation [32] or a history of wheeze or asthma-like symptoms [32], but asthma itself does not cause chronic wet cough or bronchiectasis [7]. Depending on severity, some might have a cough at baseline, but most of the children with bronchiectasis are cough free when well.

### 6.1. Cough in Bronchiectasis Exacerbations

Cough is the most reported symptom in bronchiectasis exacerbations. A recent task force report also showed that cough is the most common symptom reported in 17/21 randomized controlled trials on bronchiectasis exacerbations [58]. Bronchiectasis exacerbations are often diagnosed based on a clinician’s diagnosis of change from baseline clinical status. A robust definition of bronchiectasis exacerbation was proposed in 2012 [59]. This study proposed a combination of major and minor criteria defining bronchiectasis exacerbations. The major criteria consist of (i) significant frequency of cough (median cough score ≥ 2 [47]) over 72 h and (ii) wet cough for a minimum of 72 h. These major criteria indicate the importance of a cough during an exacerbation.

Minor criteria include sputum color (Bronkotest ≥ 3), parent/child perceived breathlessness, chest pain, crepitations, wheeze, hypoxia, elevated CRP, elevated IL6, elevated Serum Amylase A, and a raised neutrophils percentage.

Several combinations of criteria were defined as an exacerbation: one major plus one laboratory minor criteria, two major criteria, or one major criteria with any two minor criteria. Regardless of the combination of criteria, cough fulfils a central role in the definition of a bronchiectasis exacerbation [58]. There are now international consensus criteria for defining exacerbations in children and adolescents for clinical trials, and change in the cough character from dry to wet or the beginning of a new cough remains the cornerstone of identifying new exacerbation in children with bronchiectasis [58].

### 6.2. Treating Bronchiectasis Exacerbations

There are national [9] and international [60] guidelines for management of bronchiectasis exacerbations that describe the management of bronchiectasis based on four principles: (i) exacerbation management, (ii) reducing exacerbation frequency, (iii) treatment of underlying causes, and (iv) treatment of associated conditions [49]. Adequate management of acute respiratory exacerbations results in improved QoL and reduced parental stress [61]. Several studies have shown that antibiotics are effective in treating acute exacerbations [62,63,64].

### 6.3. Using Cough as an Outcome for Treating Respiratory Exacerbations in Bronchiectasis

Most often, the number of exacerbations of QoL and lung function are used as a measure of long-term bronchiectasis management. However, a recent study evaluating these endpoints reported no association between lung function, exacerbations, or other symptoms [65]. Thus, no gold standard endpoint has been defined. In the recently published pediatric RCTs, a return of cough scores to their respective baseline has been used as a sign of resolution of exacerbation [63,66].

Since cough plays a pivotal role in defining an exacerbation, it is often used as an indicator for treatment outcomes, as a new cough or change in character or severity of cough is often the main symptom during an exacerbation [67]. When treatment is initiated, monitoring the frequency of a cough, severity, and the type of cough (productive or not) provides information about treatment success [63,66]. In both of these RCTs, a non severe exacerbation was defined as an increase in cough frequency and a change in character of the cough (i.e., from dry to wet, or an increase in sputum volume or purulence) for at least three consecutive days, in combination with newer clinical findings, and resolution was defined as a return of cough to its baseline of no cough or cough character changing back from wet to dry.

## 7. Summary

PBB and bronchiectasis are paediatric respiratory conditions in which cough history is a critical diagnostic feature. In PBB, wet cough is the single symptom defining the respiratory condition. Therefore, cough resolution after appropriate duration of antibiotics treatment has now been accepted as a diagnostic criterion [16]. On the contrary, in bronchiectasis, a new cough, or change in the character of cough, defines an exacerbation. With appropriate treatment, the cough resolves, leading to the resolution of the exacerbation. Nonetheless, this does not indicate the resolution of bronchiectasis as a respiratory condition.

Despite this difference, cough reflects the clinical state of a patient in both conditions, making it an important indicator for a patients burden of disease. These common features are chronic wet/productive cough or reports of wheeze/rattles in a child who appears otherwise relatively well. Both conditions can lead to impaired cough-specific and non-cough-specific quality of life scores, which improve significantly with cough resolution. Therefore, cough is an important clinical symptom to evaluate the effectiveness of treatment in PBB, as well as bronchiectasis both clinically and as an outcome measure for clinical trials.

## Figures and Tables

**Figure 1 jcm-13-03305-f001:**
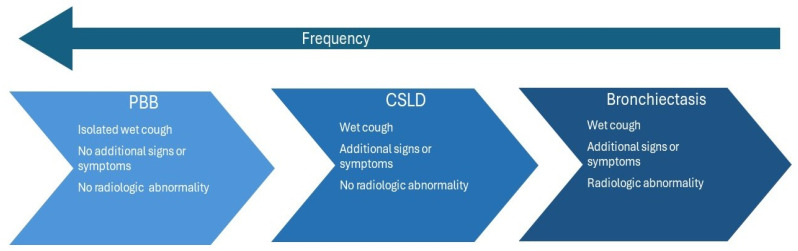
“The chronic wet cough spectrum” PBB, CSLD, and bronchiectasis represent a spectrum of chronic wet cough. All three conditions have chronic wet cough as a common clinical symptom. There is overlapping, as well as differentiating features, in these conditions on the same continuum.

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
