# Peer review of "Cough in Protracted Bacterial Bronchitis and Bronchiectasis"

_jcm, 2024, doi:10.3390/jcm13113305_

Round 1

Reviewer 1 Report

Comments and Suggestions for Authors

This is a good concise review of the topic of chronic cough.  I have two questions/suggestions related to the topic

1.      There are data, which suggest that presence of eosinophilic lower airway inflammation could be an important factor in reoccurrence of PBB.  What is the opinion of authors on overlap of PBB/CSLD/bronchiectasis with asthma-like conditions?

2.      What is the role of inflammatory markers in the diagnosis of PBB?  It is important to emphasize that elevated blood inflammatory markers are not required  for the diagnosis of PBB.  It is my experience that the practitioners get confused with this and often deny possibility of lower airway bacterial inflammation because WBC and CRP are normal. 

There are some suggested corrections:

1.      The authors state: “children with CSLD present with additional signs and/or symptoms  in the presence of a normal c-HRCT”.  This is not accurate.  Children with CSLD usually have abnormal CT scans, which reflect certain degree of bronchial wall inflammation without bronchiectasis 10.1007/s00431-016-2743-5   

2.      Figure 1.  Both CSLD and bronchiectasis are marked as “no radiologic abnormality”, which is not correct and confusing

3.      Repeating the same reference, please find the way to do it once in the beginning or in the end of the sentence (ref 7 and ref 13)

4.      The treatment of choice for PBB is amoxicillin-clavulanate, because of its activity against 147 β-lactamase being effective against the most frequently encountered pathogens; H. influ- 148 enzae, S. pneumoniae, and M. Catarrhalis (15).  This is accurate only with regards to Heamophilus and Moraxella, but resistance of Strep pneumoniae is due to penicillin binding proteins (PBP) abnormalities.  ref doi: 10.1016/j.micres.2022.127221

5.      Reference 47 is “implanted” in the text but it is not quite clear what it means.  My guess is that it is related to definition of cough score, but it needs to be explained better. 

Author Response

This is a good concise review of the topic of chronic cough.  I have two questions/suggestions related to the topic.

Response: Thank you for the kind words.

  1. There are data, which suggest that presence of eosinophilic lower airway inflammation could be an important factor in reoccurrence of PBB.  What is the opinion of authors on the overlap of PBB/CSLD/bronchiectasis with asthma-like conditions?

 Response: Thank you for this comment. The ERS statement on PBB mentions “No airway eosinophilia was observed in any study and a single study described an increase in the percentage of lymphocyte” [1] Similar observations were made in a review paper in 2020 [2]. There are data from endobronchial biopsies which have demonstrated presence of eosinophils in children with wet cough for more than 8 weeks, these are data from biopsy and not BAL. Furthermore, this study did not define eosinophilia and have reported absolute number of different cells. [3]

However, we agree with the reviewer that there is an overlap in some children with asthma-like condition and hence have added this to the revised manuscript stating: This study also demonstrated an overlap with asthma-like condition; with clinician-diagnosed asthma at final follow-up was present in 27.1% of children with PBB. ( Page 4, lines 148-150)

  1. What is the role of inflammatory markers in the diagnosis of PBB?  It is important to emphasize that elevated blood inflammatory markers are not required for the diagnosis of PBB.  It is my experience that the practitioners get confused with this and often deny possibility of lower airway bacterial inflammation because WBC and CRP are normal. 

Response: The inflammatory markers in PBB is still an area of ongoing research. While there are data on increased levels of BAL interleukin (IL)-8, active matrix metalloproteinase-9 and IL-1β which correlate with the degree of neutrophilia, there are no systemic markers identified yet.  BAL fluid of PBB patients can have higher α-defensin, IL-1 pathway cytokines, and CXCR2 gene and protein expression too. [4-7] However these markers are for research purposes only and not clinically-used markers of inflammation which, as stated, are normal in PBB. We have now added a sentence to reflect this in the manuscript Children with PBB have preserved systemic adaptive immunity with normal serum immunoglobulin levels and normal antibody-mediated responses to both protein (tetanus) and conjugated protein-polysaccharide based (H. influenzae type b) vaccines, and do not have any other signs of systemic inflammation”(page 3, lines 83-86)

There are some suggested corrections:

  1. The authors state: “children with CSLD present with additional signs and/or symptoms in the presence of a normal c-HRCT”.This is not accurate.  Children with CSLD usually have abnormal CT scans, which reflect certain degree of bronchial wall inflammation without bronchiectasis 10.1007/s00431-016-2743-5.    

 Response: We agree and have now changed the sentence to” whereas children with CSLD present with additional signs and/or symptoms in the absence of bronchial dilatation on c-HRCT” for clarity (page 2, line 52)

  1. Figure 1.  Both CSLD and bronchiectasis are marked as “no radiologic abnormality”,which is not correct and confusing.

 Response: Thank you for pointing out this typing mistake. We have now corrected it.

  1. Repeating the same reference, please find the way to do it once in the beginning or in the end of the sentence (ref 7 and ref 13)

 Response: Thank you for this suggestion. While we agree with the reviewer, we felt that the reference were repeated at the end of the sentence in the same topic and not in the same sentence.

  1. The treatment of choice for PBB is amoxicillin-clavulanate, because of its activity against 147 β-lactamase being effective against the most frequently encountered pathogens; H. influ- 148 enzae, S. pneumoniae, and M. Catarrhalis (15). This is accurate only with regards to Heamophilus and Moraxella, but resistance of Strep pneumoniae is due to penicillin binding proteins (PBP) abnormalities.  ref doi: 10.1016/j.micres.2022.127221

 Response: We agree with the reviewer and have change the sentence to “The treatment of choice for PBB is amoxicillin-clavulanate, because of its activity against β-lactamase and being effective against the most frequently encountered pathogens; H. influenzae, S. pneumoniae, and M. Catarrhalis”(page 4, line 157)

  1. Reference 47 is “implanted” in the text but it is not quite clear what it means.  My guess is that it is related to definition of cough score, but it needs to be explained better. 

 Response:  The reference is indeed related to the cough score. We have now changed the position of this reference in the sentence to make it clearer. “The cough score in children [47] has been validated as a clinical and research tool and resolution of cough identified as a marker of improvement in cough.[40, 47] The cough score is a verbal category descriptive score (VCD) ranging from 0 (no cough) to 5 (cannot perform most usual activities due to severe coughing).[47]”

Reviewer 2 Report

Comments and Suggestions for Authors

A little more detail on management of bronchiectasis.

Comments on the Quality of English Language

Good.

Author Response

Thank you for this suggestion. Since this review is focused on Cough in PBB and Bronchiectasis, we have provided references to the national and international guidelines on the management of bronchiectasis in section 6.2. We feel that providing more details on the management of bronchiectasis will be out of context here. 

Reviewer 3 Report

Comments and Suggestions for Authors

This is a comprehensible review on a spectrum of condition presenting with chronic cough in children. The authors suggested a systematic approach of the chronic wet cough spectrum. 

Regarding the structure and contents of the manuscript I thought it appropriate as it is. However several minor points should be checked.

line 54, (BARThe BAR is ...

line 130/131, duration of productive cough had a significant negative correlation (r=-0.51) with airway obstruction ..? Do you mean that longer period is associated (LESS or MORE) severe obstruction; quite confusing.

line 149 "An" study

line 200 features "then" those with

line 221 elevated "SAA"

line 270 cough-specific and "generic" quality of life scores 

Comments on the Quality of English Language

Mostly the authors Quality of English is excellent. I could easily understand what the authors mean to say.

Author Response

This is a comprehensible review on a spectrum of condition presenting with chronic cough in children. The authors suggested a systematic approach of the chronic wet cough spectrum. 

Response: Thank you to the reviewer for the kind words.

Regarding the structure and contents of the manuscript I thought it appropriate as it is. However, several minor points should be checked.

line 54, (BARThe BAR is ...

Response: Thank you for pointing out this typing mistake. We have now corrected it.

line 130/131, duration of productive cough had a significant negative correlation (r=-0.51) with airway obstruction ..? Do you mean that longer period is associated (LESS or MORE) severe obstruction; quite confusing.

Response: Thank you. We have now reworded the sentence now to “. In adult non-smokers newly diagnosed with bronchiectasis, the duration of productive cough had a significant negative correlation with airway obstruction measured on spirometry (r= -0.51% predicted FEV1 per year of chronic cough, 95% CI −0.69, −0.33; p < 0.001), demonstrating that the symptoms start in childhood” to make it clearer to read (page 4, lines 137-138)

line 149 "An" study

Response: Thank you for pointing out this typing mistake. We have now corrected it.

line 200 features "then" those with

Response: Thank you for pointing out this typing mistake. We have now corrected it to than.

line 221 elevated "SAA"

Response: Thank you for pointing out this typing mistake. We have now explained this abbreviation as Serum Amylase A.

line 270 cough-specific and "generic" quality of life scores 

Response: Thank you for pointing this out. We have now changed generic to “non-cough specific”.
